# Equilibrium Bond Lengths from Orbital-Free Density Functional Theory

**DOI:** 10.3390/molecules25081771

**Published:** 2020-04-13

**Authors:** Kati Finzel

**Affiliations:** Faculty of Chemistry and Food Chemistry, Technische Universität Dresden, Bergstraße 66c, 01069 Dresden, Germany; Kati.Finzel@tu-dresden.de

**Keywords:** orbital-free density functional theory, bifunctional, Pauli potential, Pauli kinetic energy, chemical bonding, real space, atomic fragment approach

## Abstract

This work presents an investigation to model chemical bonding in various dimers based on the atomic fragment approach. The atomic fragment approach is an ab-initio, parameter-free implementation of orbital-free density functional theory which is based on the bifunctional formalism, i.e., it uses both the density and the Pauli potential as two separate variables. While providing the exact Kohn-Sham Pauli kinetic energy when the orbital-based Kohn-Sham data are used, the bifunctional formalism allows for approximations of the functional derivative which are orbital-free. In its first implementation, the atomic fragment approach uses atoms in their ground state to model the Pauli potential. Here, it is tested how artificial closed-shell fragments with non-integer electron occupation perform regarding the prediction of bond lengths of diatomics. Such fragments can sometimes mimic the electronic structure of a molecule better than groundstate fragments. It is found that bond lengths may indeed be considerably improved in some of the tested diatomics, in accord with predictions based on the electronic structure.

## 1. Introduction

Orbital-free density functional theory (OF-DFT) promises a reliable physical description of matter on the basis of quantum mechanics [1], while staying at a moderate level of computational costs [2,3,4,5,6,7]. The enormous computational speed-up is due to the fact that, in OF-DFT a single Euler equation directly determines the electron density of the system. This is in contrast to the well-known, orbital-based Kohn-Sham (KS) method [1,8], where a system of *N* non-interacting electrons has to be solved. However, OF-DFT requires finding sufficiently accurate approximations for the kinetic energy, being a considerable challenge for theoretical physicists and chemists [9].

In the early days, density functional approximations were mostly based on conventional gradient expansions [10,11,12,13,14,15,16,17,18,19,20,21]. Unfortunately, those expansions do not form a convergent series [22], and thus, this route was abandoned. Later those attempts were extended to generalized-gradient-approximation techniques, where the explicit functional approximation is motivated by conjoint arguments [23,24] or the fulfillment of additional constraints [9,25,26]. Besides GGA-type functionals, kinetic energy expansions based on moment densities were studied [27], as well as functionals based on response theory [28,29], parameterized power series expansions [30,31,32], and information-theory motivated expressions [33,34,35]. Following the ideas of March, it would be sufficient to model the Pauli kinetic energy [36], as it constitutes the only unknown functional part of the kinetic energy, while the remaining part, the so-called von Weizsäcker kinetic energy [12] is known analytically in terms of the electron density. Consequently, the Pauli kinetic energy and the corresponding Pauli potential [36,37,38,39,40,41,42,43,44,45,46] have been subject for intense studies over decades.

Following the ideas of Hohenberg and Kohn [1], one usually aims to approximate the energy functional as an integral over a kernel function that analytically depends on the electron density and possibly other ingredients. Another route for the design of energy expressions is based on the so-called non-analytical functionals [47,48,49,50,51]. This approach exploits the homogeneous scaling behavior of a given density functional [52], while suppressing its implicit density dependence in the functional derivative. Consequently, this approach deals with two separate variables, namely the potential and the electron density. The reformulation is exact for the exact density-potential pair, however, it allows for further approximations of the formal functional derivative. This leads to a systematic route of functional approximations, that belong to the group of ab-initio, parameter-free implementations of orbital-free density functional theory.

This work further exploits the proposed formalism. While the first order atomic fragment approach involves atomic fragments in their groundstate, the recent work investigates the influence of the fragment Pauli potential on the equilibrium bond lengths using groundstate and closed-shell atomic fragments. Closed-shell atoms are closer to the real space fragment within a molecule and as such, the corresponding fragment Pauli potential mimics to some extend the relaxation of the molecular Pauli potential during the process of molecular formation. It is shown that this relaxation presents an eminent part for an orbital-free description of chemical bonding.

## 2. Theory

According to the Hohenberg-Kohn (HK) theorems [1], the total energy *E* of a system can be expressed as a functional of the electron density ρ:
(1)$$E\left[\rho \right]=T_{s}\left[\rho \right]+V_{ee}\left[\rho \right]+V_{Z}\left[\rho \right]+V_{NN}\;\;.$$

Here, $T_{s}\left[\rho \right]$ is the non-interacting kinetic energy, $V_{ee}\left[\rho \right]$ is the Coulomb repulsion between the electrons, $V_{Z}\left[\rho \right]$ is the electron-nuclear attraction energy, and $V_{NN}$ is the repulsion energy between the nuclei.

Usually, $T_{s}\left[\rho \right]$ and $V_{ee}\left[\rho \right]$ are further decomposed into meaningful parts. The non-interacting kinetic energy can be regarded as being constructed from a bosonic part, the von Weizsäcker term [12] $T_{W}$, and a remainder, called Pauli kinetic energy $T_{P}$, which consequently, is defined as the difference: [36]
(2)$$T_{P}=T_{s}-T_{W}\;\;.$$

The von Weizsäcker kinetic energy $T_{W}$ can be expressed in form of an analytical density functional, given by [12,36]:(3)$$T_{W}\left[\rho \right]=\int t_{W}\left(\vec{r}\right)\;d\vec{r}=\int \frac{1}{8}\frac{{\left[\vec{\nabla }\rho \left(\vec{r}\right)\right]}^{2}}{\rho (\vec{r})}\;d\vec{r}\;\;,$$
where the integral kernel $t_{W}\left(\vec{r}\right)$ is called Weizsäcker kinetic energy density.

The electron-electron repulsion $V_{ee}\left[\rho \right]$ is usually split into the Hartree energy $E_{H}\left[\rho \right]$ and the exchange-correlation energy $E_{XC}\left[\rho \right]$, where the Hartree energy is given by:(4)$$E_{H}\left[\rho \right]=\frac{1}{2}\int \int \frac{\rho \left(\vec{r}\right)\rho \left({\vec{r}}^{\prime }\right)}{|\vec{r}-{\vec{r}}^{\prime }|}\;d{\vec{r}}^{\prime }\;d\vec{r}\;\;$$
and the exchange-correlation energy $E_{XC}\left[\rho \right]$, here, is approximated by the exchange-only local density approximation $E_{X}^{\mathrm{LDA}}\left[\rho \right]$ for simplicity (since the focus is set on approximating the kinetic energy):(5)$$\begin{array}{c} E_{X}^{\mathrm{LDA}}\left[\rho \right]=-C_{X}\int \rho^{\frac{4}{3}}\left(\vec{r}\right)\;d\vec{r}\;\; \end{array}$$
with $C_{X}=3/\left(4\pi \right){\left(3\pi^{2}\right)}^{1/3}\approx 0.73856$.

The electron nuclear attraction energy is known exactly as electron density functional:(6)$$V_{Z}\left[\rho \right]=\int \rho \left(\vec{r}\right)v_{Z}\left(\vec{r}\right)\;d\vec{r}\;\;$$
where the nuclear potential of a molecule is given by the superposition $v_{Z}\left(\vec{r}\right)=\sum_{A}v_{Z}^{A}\left(\vec{r}\right)$ of all atomic nuclear potentials $v_{Z}^{A}\left(\vec{r}\right)=-Z_{A}/\left|\vec{r}-{\vec{R}}_{A}\right|$, with $Z_{A}$ being the nuclear charge and ${\vec{R}}_{A}$ the nuclear coordinates.

The nuclear repulsion energy is evaluated classically as:(7)$$V_{NN}=\sum_{A}\sum_{B=A+1}\frac{Z_{A}Z_{B}}{R_{AB}}\;\;$$
where the sum runs over all nuclei and $R_{AB}$ is the respective distance between the two nuclei. Thus, the Pauli kinetic energy $T_{P}$ remains the only unknown term in order to achieve an orbital-free description of the molecule.

In this work, the Pauli kinetic energy is evaluated from the so-called bifunctional expression [37,48,51]:(8)$$T_{P}\left[\rho ,v_{P}\right]=-\frac{1}{2}\int \rho \left(\vec{r}\right)\;\vec{r}\cdot \vec{\nabla }v_{P}\left(\vec{r}\right)\;d\vec{r}\;\;\;$$
involving both: the electron density $\rho (\vec{r})$ and the Pauli potential $v_{P}\left(\vec{r}\right)$ as separate variables. The bifunctional expression exploits the homogenous scaling behavior [52] of the otherwise unknown functional expression. Consequently, the bifunctional expression is exact, when the molecular electron density and the molecular Pauli potential are inserted into Equation (Equation 8). The Pauli potential of the molecule is only known in terms of the KS eigenfunctions $\phi_{i}\left(\vec{r}\right)$ and respective eigenvalues $\epsilon_{i}$ for the molecule of interest: [37]
(9)$$v_{P}\left(\vec{r}\right)=\frac{\delta T_{p}}{\delta \rho }=\frac{t_{P}\left(\vec{r}\right)}{\rho (\vec{r})}+\sum_{i}\left(\epsilon_{M}-\epsilon_{i}\right)\frac{|\phi_{i}\left(\vec{r}\right){|}^{2}}{\rho (\vec{r})}\;\;\;$$
where the sum runs over all occupied eigenfunctions, $\epsilon_{M}$ is the highest occupied eigenvalue of the system, and the Pauli kinetic energy $t_{P}\left(\vec{r}\right)$ is given by:(10)$$t_{P}\left(\vec{r}\right)=\frac{1}{2}\sum_{i}{\left|\nabla \phi_{i}\left(\vec{r}\right)\right|}^{2}-t_{W}\left(\vec{r}\right)\;\;\;.$$

Therefore, within an orbital-free formalism the molecular Pauli potential must be subjected to further approximations.

Recently the atomic fragment approximation has been introduced. In its simplest form, the so-called zeroth order approximation [48], all energy terms are treated within the atomic fragment approach keeping fragments frozen. The molecular density is approximated by the promolecular density and all functional derivatives, the potentials, are given as simple superposition of their atomic counterparts. In detail this means, that atoms in their groundstate are calculated with a standard electronic structure program and the atomic KS data from the atoms in their groundstate directly serve to compute the energies without any optimization procedure. The total energy is evaluated subsequently from the corresponding bifunctional expression. This very simplistic treatment of a molecule already captures the main characteristics of chemical bonding, namely the dependence of the total energy on the internuclear distance. However, equilibrium bond distances deserve amelioration. The first order fragment approach [51] allows for a variational relaxation of the electron density, and thus, for a better description of chemical bonding in molecules. In the first order variant, the total energy of a system is given by:(11)$$\begin{array}{c} E\left[\rho ,v_{P}\right]=T_{W}\left[\rho \right]+T_{P}^{\Omega }\left[\rho ,v_{P}^{\Omega }\right]+E_{H}\left[\rho \right]+E_{X}^{\mathrm{LDA}}\left[\rho \right]+V_{Z}\left[\rho \right]+V_{NN}\;\;. \end{array}$$

Thus, in the first order fragment approach only the Pauli kinetic energy is expressed within the bifunctional formalism (notice, that for analytically known density functionals, the density functional and the bifunctional expression yield exactly the same value):(12)$$T_{P}^{\Omega }\left[\rho ,v_{P}^{\Omega }\right]=-\frac{1}{2}\int \rho \left(\vec{r}\right)\;\vec{r}\cdot \vec{\nabla }v_{P}^{\Omega }\left(\vec{r}\right)\;d\vec{r}\;\;\;$$
and only the molecular Pauli potential is approximated by its atomic fragment variant:(13)$$v_{P}^{\Omega }\left(\vec{r}\right)=\sum_{A}v_{P}^{A}\left(\vec{r}-{\vec{R}}_{A}\right)\;\;\;.$$

In the first order fragment approach, the atomic Pauli potentials were taken from atoms in their electronic groundstate (gs) configuration [51]. The largest influence of the electronic state of the atomic fragments is on the atomic Pauli potentials and as such on the fragment Pauli potential of the molecule. A much smaller indirect influence comes via the molecular density, which is also build from the atomic fragments, see Section 4 for further details. While the electron density itself is not altered by the electronic fragment state, there is a slight influence on the total energy, due to those energy parts that are treated in spin-polarized formalism. The atomic fragment approach allows for a relaxation of the electron density expressed as the sum over nodeless, atom-centered Slater functions, see Section 4 for details, by adjusting the exponents of the Slater functions, while keeping their occupations fixed. In this work, the influence of the electronic fragment states on the equilibrium bond length is studied in detail. In the present work the molecular electron density is build from closed-shell fragments (with equal number of alpha and beta electrons, leading to fractional occupations for uneven number of total electrons in the fragment), while the influence of the Pauli potential is explored systematically by taking either the atomic Pauli potential (PP) from atoms in their groundstate (gs) or from atoms in an artificial closed-shell configuration (cs). Consequently, the molecular Pauli potential is approximated by the superposition of closed-shell (cs) or groundstate (gs) atomic fragments in order to study which fragments perform better with respect to chemical bonding. The electronic density as well as all energetic contributions are treated within closed-shell formalism.

Notice that the approximations done here are due to physically meaningful considerations about the molecular formation process. At this level of theory, no approximate analytical functional expression, nor parameters or fitting procedures have been introduced. Furthermore, the atomic fragment approach captures—by construction—important characteristics of the Pauli potential, e.g., it maintains the atomic shell structure in the core region and is non-negative everywhere in space. The fragment Pauli potential is constructed from the sum of the individual atomic KS Pauli potentials which obey the non-negativity condition. Consequently, the fragment Pauli potential is non-negative everywhere. The non-negativety of the Pauli potential has been shown to be an important aspect for the design of orbital-free approximations for the kinetic energy [9,26,53].

## 3. Results and Discussion

The proposed procedure was applied to various dimers. Total energy curves were calculated by optimizing the valence density and the equilibrium bond distances were evaluated from the corresponding minima. The data are compiled in Table 1 together with the corresponding KS data and experimental values. Relative errors with respect to experimental bond lengths are also given in the table. For the examined molecules the orbital-based KS approach, see the second and third column, yielded an appropriate description of chemical bonding with respect to the equilibrium bond lengths. The KS bond distances were slightly longer compared to their corresponding experimental data and relative errors were roughly 1% for all examined cases, except for the C${}_{2}$ molecule, where the error was about 13%. Thus, a mean absolute percentage error (MAPE) of a few percent was the target for an orbital-free approach. The last eight columns contained bond lengths from orbital-free calculations. While all molecules were bound within the zeroth order approach [48], the corresponding bond lengths were in general too long, yielding a MAPE of 24.9% for the test set N${}_{2}$, O${}_{2}$, CO, and Be${}_{2}$. The zeroth order level neither allowed for the relaxation of the Pauli potential nor of the density. The latter shortcoming was overcome by the first order atomic fragment approach, in which the electron density was energetically minimized by optimizing the exponents of simple nodeless Slater functions describing the core and the valence electron density, respectively. Here, it was sufficient to optimize the exponent governing the valence region, as optimization of the core density does not alter bond lengths [51]. The results of the first order approach are compiled in the sixth and seventh column. As can be seen from the data bond length significantly improved at the first order level, yielding a MAPE of 10.8% for the molecules N${}_{2}$, O${}_{2}$, CO, and Be${}_{2}$.

While the density was optimized at first order level, the molecular Pauli potential, built from the superposition of the corresponding atomic fragments, was kept frozen for a given internuclear distance. The strength of the Pauli repulsion is crucial for an appropriate description of chemical bonding and consequently, determines the equilibrium structure. Therefore, a good model of the Pauli potential must somehow account for the processes that take part during the molecular formation. While atomic groundstate configurations are usually high-spin configurations due to the Pauli principle, within a molecule all valence electrons of the respective atomic fragments are shared among each other [57] and thus, a closed-shell state results (in most cases). This electronic reorganization requires a process called promotion, in which the atomic fragment undergoes to a state of higher energy (compared to its groundstate) that allows for an effective sharing of electrons within the molecule [57]. Therefore, within the molecule itself, the atomic fragment feels an electronic surrounding that corresponds more to a closed shell configuration than to its proper atomic groundstate. This aspect has been considered in the present work. While at first order level the molecular Pauli potential is evaluated as the superposition of atomic Pauli potentials from atoms in their groundstate, the effects on the Pauli potential due to the molecular environment can be mimicked by approximating the molecular Pauli potential as the superposition of closed-shell atomic fragments.

The influence of respective fragments on the approximate molecular Pauli potential and consequently, on the equilibrium bond distance can be studied by comparing columns 8 and 10 together with their respective relative errors. The current method employing groundstate fragments (gs PP) was close to the first order fragment approach. Differences only occurred for spin-polarized energy contributions and thus, bond distances did not alter much. The MAPE of the set test N${}_{2}$, O${}_{2}$, CO, and Be${}_{2}$ was with 9.7% relatively close to the MAPE of 10.8% from the first order level. In contrast, employing closed-shell fragments further reduced errors in bond lengths, yielding a MAPE of 6% over the set of tested molecules compared to a MAPE of 8.2% when groundstate fragments are used. Surely, some of the test molecules, in particular O${}_{2}$ and Be${}_{2}$, did not show an amelioration of the equilibrium bond distance and further work remains in the field of OF-DFT. Nevertheless, the present study suggests that the closed-shell variant is a much better starting point for further systematic approaches in the atomic fragment approach.

Employing closed-shell fragments, thus, yields a better description of chemical bonding compared to the performance with groundstate fragments. As mentioned before, this is due to the reorganization of electrons during the process of molecular binding. Whereas groundstate atoms and their corresponding atomic fragment Pauli potentials are good estimates for a nearly dissociated system, the atomic fragments within a molecule close to its equilibrium structure usually resembles closed-shell atoms. For this reason, the present approach ameliorates with respect to bond distances. The influence of the atomic configuration on the resultant molecular Pauli potential, thus, plays the mayor role for an appropriate description of chemical bonding in OF-DFT.

In order to reveal the role of the Pauli potential on the process of chemical bonding and the resultant equilibrium distances in more detail, the molecular KS Pauli potential (evaluated from KS orbitals) for the nitrogen molecule at equilibrium bond length has been depicted in Figure 1 together with the Pauli potentials from the first order atomic fragment approach and the present work. Here, the KS data are depicted in the second column, while the first and third column show the Pauli potential for the orbital-free approach of the recent approach and the first order level, respectively. The first and second row of Figure 1 display the relative height field and the orthoslice of the Pauli potential along the molecular axis, respectively. The latter reveals how close both fragment models are compared to the orbital-based KS object. All orthoslices, shown in Figure 1e–g, display almost spherical behavior around the atomic nuclei and fall off rapidly. As a consequence the Pauli potentials are similar even in the bonding region. Only a slight dip along the molecular axis, see the KS Pauli potential in Figure 1b, is missing in the fragment approaches. Otherwise the form and absolute values are well preserved by the suggested fragment approximation. This is due to the fact that the Pauli potential is relevant only in the core region [58,59,60], which remains unaltered during the process of chemical bonding. At the shell boundary between the first and the second shell the Pauli potential exhibits a maximum in order to induce the typical shell structure in the radial electron density [58] and falls off rapidly in the valence region. For this reason a simple atomic superposition is a good estimate for the total molecular entity. The closed-shell fragment potential is closer to the KS Pauli potential than its gs counterpart, cf. Figure 1i,k, depicting the difference between the KS Pauli potential and the respective fragment approach. This is due to the size of the respective cones depicted in the first row of Figure 1. The KS data and the closed-shell fragment Pauli potential are somewhat smaller in size than the Pauli potential obtained from groundstate atoms. For a better comparison between both models, the difference between the closed shell data and the groundstate model is displayed in Figure 1j. As can be seen from the figure, the closed-shell model yields thinner cones for the Pauli potential compared to the data obtained from atomic groundstates. For this reason the difference to the KS Pauli potential is considerably smaller when closed-shell fragments are employed than for the first order level, cf. Figure 1i,k, respectively. Consequently, the closed-shell model yields a better agreement with experimental bond lengths than the first order atomic fragment approach.

Besides reasonable equilibrium bond lengths, the current approach is able to properly model important energetic characteristics of chemical binding curves [61]. Exemplarily the binding curve for N${}_{2}$ is depicted in Figure 2 together with its kinetic and potential components. Notice that all displayed terms have been evaluated as the difference between the molecular and the (gs) atomic values. The inset of Figure 2 displays the binding energy in the region close to the equilibrium distance, showing the position of the energetic minimum and the form of the binding curve. The outer figure depicts the binding curve together with its kinetic and potential components over a large range of internuclear distances in order to reveal the subtle energetic balance between the two, that is necessary to achieve chemical binding. First of all, notice how shallow the energetic minimum of the binding curve was compared to the magnitude of energetic values of the molecule and its atomic fragments. This problem is sometimes referred as weighting the captain on its ship as well as the ship alone in order to determine the captains weight [62]. Since the kinetic energy is similar in magnitude as the total energy itself, an appropriate model of the Pauli potential is crucial for the description of the energetic binding curve. Notice that the molecular contribution of the von Weizsäcker term, shown in green, was negative during the process of chemical bonding. Thus, the increase of the kinetic energy during the formation of the molecule is an aspect of the Pauli kinetic energy alone. Figure 2 depicts the closed-shell fragment Pauli kinetic energy, shown by the orange line, as well as the total kinetic energy, shown by the red line. At large distances the interatomic Pauli repulsion was roughly zero and the total kinetic energy followed the von Weizsäcker term. This resulted in a slight decrease of values for the kinetic energy. However, the interatomic Pauli repulsion started to significantly increase as the atoms got closer, while the von Weizsäcker term moderately decreased. As a consequence, the total kinetic energy followed the Pauli kinetic energy for moderate and small internuclear distances. The figure also illustrates the breakdown of the atomic fragment approach at very short internuclear distances, here, around 0.5 Bohr. This corresponds to those internuclear distances, when the two core regions started to overlap. Recall that the valence-core shell boundary for the nitrogen atom evaluated with the help of the electron-localization function is 0.47 Bohr [63]. Fortunately, chemical bonding takes place far away from this region of internuclear distances as only valence regions considerably overlap.

One important aspect in the theory of chemical bonding is the relationship between the kinetic and the potential energy curves [61]. As can be seen from the figure, the kinetic energy curve passed a minimum at 3.22 Bohr, indicated by the red dashed line. As explained in the previous paragraph, the minimum is due to the interplay of the decreasing von Weizsäcker contribution and the (at small internuclear distances) increasing Pauli repulsion. The kinetic energy exhibits its minimum at large internuclear distance. The total energy, however, still decreases, because the contribution from the potential energy curve, shown in blue, still decreases. The potential curve exhibited its minimum, displayed by the blue dashed line, at small internuclear distances, here, at 1.20 Bohr. Since the potential curve decreased faster than the increase of the internuclear Pauli repulsion, the chemical bond was formed in the intermediate region. The energetic minimum for the nitrogen molecule, shown by the black dashed line, was found at 2.12 Bohr.

The correct behavior of the kinetic versus potential curves is an important aspect in the theory of chemical bonding [61]. The chemical bond is formed by paying a small price due to the increase of the kinetic energy, while gaining the (larger) energetic contribution from the potential part. Usually, this behavior is attributed to the non-classical interference terms from wavefunction-based electronic structure theory. In this work, it is shown that those aspects of chemical bonding can equally be achieved with orbital-free density functional theory.

## 4. Materials and Methods

Energy minimization was performed by our own code using atom-centered, squared, real-type nodeless Slater functions (1S, 2S) to construct the electron density:
(14)$$\rho \left(\vec{r}\right)=\sum_{i}g_{i}\phi_{i}^{2}\left(\vec{r}-{\vec{R}}_{A}\right)\;\;.$$
where the occupations $g_{i}$ correspond to the total number of electrons in the corresponding shell. Nodeless spherical Slater functions are given by [64,65]:
(15)$$\phi_{i}\left(\vec{r}\right)=N_{0}r^{a_{i}}e^{-\alpha_{i}r}\;\;$$
with:(16)$$N_{0}=\sqrt{\frac{{\left(2\alpha_{i}\right)}^{2a_{i}+3}}{4\pi (2a_{i}+2)!}}\;\;,$$
$a_{i}=n^{*}-1$ and $\alpha_{i}=\left(Z-s\right)/n^{*}$, where *Z* is the nuclear charge, $n^{*}$ is an effective quantum number, and *s* is the so-called shielding constant [65]. Shell concept, occupation and shielding constants are determined according to the Slater rules [65], e.g., second row elements have one shell in the core region and one single shell in the valence region (grouping s- and p-electrons together). However, the atomic valence electron density was restricted to closed-shell states in order to model the closed-shell state for the molecule. Atomic closed-shell states were obtained by equally filling up alpha and beta electrons until the total number of electrons was reached. Consequently, closed-shell atoms had fractional occupations for an uneven number of total electrons. The Slater exponents for the valence density, here $\alpha_{2S}$ in all cases, were projected to an optimization procedure in order to minimize the total energy. Bear in mind that optimization of exponents with respect to the energy leads to a system of non-linear equations. Therefore, the minimization procedure had to be performed iteratively.

The current approach allows for the relaxation of the density by adjusting the exponents $\alpha_{i}$. The Pauli potential, however, obtained by superposition of atomic closed-shell atomic Pauli potentials or groundstate atoms, respectively, was kept fixed (at a given internuclear distance) during the optimization process. Those atomic Pauli potentials were taken from an LDA (Xonly) KS calculation performed with ADF [54] using atomic closed-shell configurations and the QZ4P basis sets.

## 5. Conclusions

This work presents an ab-initio, parameter-free, orbital-free implementation of density functional theory. In the present approach the functional value of the Pauli kinetic energy is evaluated from the density-potential pair using the bifunctional formalism. The bifunctional formalism allows to treat the functional derivative, the potential, and the electron density as two separate variables, while yielding exactly the Kohn-Sham Pauli kinetic energy when the corresponding orbital-based Kohn-Sham density and Pauli potential are inserted. However, the bifunctional approach allows for further approximations of the formal functional derivative, e.g., the atomic fragment approach. Within the atomic fragment approach, the molecular Pauli potential is approximated by the superposition of atomic Pauli potentials. While in previous attempts only atomic groundstate fragments have been employed, this work exploits the influence of the atomic fragment states on the ability to model chemical bonding. Thus, in the present work the molecular Pauli potential is approximated by the superposition of closed-shell and groundstate atomic Pauli potentials, respectively, and the electron density is determined by energy minimization. The resulting energy curves were evaluated for various dimers in order to determine the respective equilibrium bond lengths. It was shown that the corresponding bond distances improve for most diatomics if closed-shell fragments are used instead of groundstate fragments, with considerable improvements for some of the considered molecules and with a MAPE of only 6%.

This improvement is due to the incorporation of the effects causes by the electronic reorganization in the Pauli potential due to the process of molecular binding. Precisely, due to the electronic promotion and electron sharing in the molecule, the respective atomic fragments resembles more closed-shell atoms than atoms in their groundstate (first order approximation). Consequently, closed-shell fragments yield a significantly better description of chemical bonding for some of the investigated diatomics.

## Figures and Tables

**Figure 1 molecules-25-01771-f001:**
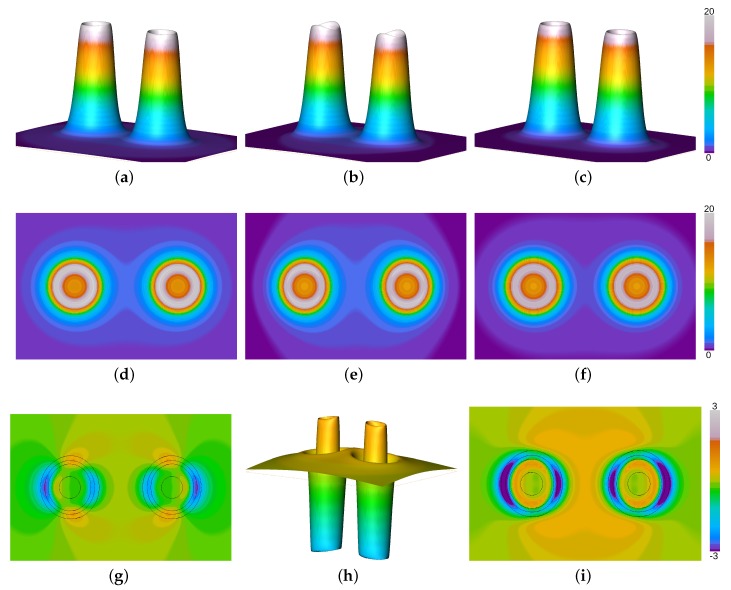
Pauli potentials for the N2 molecule from the atomic fragment approach using closed shell fragments (first column) together with the first order level (third column) and KS data (second column). (**a**) relative height field using closed-shell (cs) atomic fragments, (**b**) relative height field using molecular Kohn-Sham (KS) orbitals, (**c**) relative height field using groundstate (gs) atomic fragments, (**d**) orthoslice using cs atomic fragments, (**e**) orthoslice using molecular KS orbitals, (**f**) orthoslice using gs atomic fragments, (**g**) difference between the molecular and the cs Pauli potential (in black: isolines of the KS Pauli potential indicating regions of high contribution to the Pauli kinetic energy), (**h**) difference between the cs and the gs Pauli potential, (**i**) difference between the molecular and the gs Pauli potential (in black: isolines of the KS Pauli potential indicating regions of high contribution to the Pauli kinetic energy).

**Figure 2 molecules-25-01771-f002:**
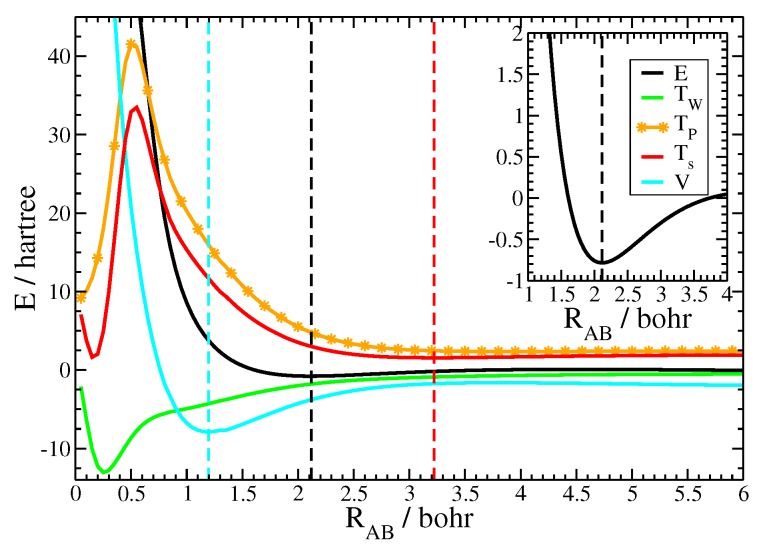
Energetic binding curve for N${}_{2}$ together with its kinetic and potential components. All data are evaluated as the difference between the molecular data and the groundstate atoms. Black: binding energy *E*, red: total kinetic energy $T_{s}$, orange: Pauli kinetic energy $T_{P}$, green: von Weizsäcker kinetic energy $T_{W}$, blue: total potential energy *V*. The dashed lines mark the minima of the respective curves. Inset: binding energy *E* around the equilibrium distance.

**Table 1 molecules-25-01771-t001:** Bond distances (in Bohr) obtained from experiment, Kohn-Sham (KS) data and orbital-free density functional theory (OF-DFT) using the atomic fragment approach of zeroth and first order level, together with the bond distances from the recent work using the atomic groundstate fragments (gs PP) and closed-shell atomic fragments (cs PP) in order to approximate the molecular Pauli potential, respectively. The respective relative errors to the experimental data δ are given in the subsequent columns. Their absolute average, the mean absolute percentage error (MAPE) is given in the last row. The zeroth order level has been performed with frozen fragments. Equilibrium bond length from first order level and the recent work are evaluated at valence optimized level [51]. Bond distances from KS data are obtained with ADF [54] using LDA(Xonly) level with the QZ4P basis sets.

				OF-DFT							
								Recent Work			
	Exp. [55,56]	KS	δ	0th [48]	δ	1th [51]	δ	gs PP	δ	cs PP	δ
N${}_{2}$	2.07	2.09	0.8	2.9	39.8	2.30	10.9	2.26	8.7	2.12	2.1
O${}_{2}$	2.28	2.31	1.0	2.6	13.9	1.85	−18.9	1.84	−19.1	1.83	−19.6
CO	2.13	2.15	0.9	3.0	40.7	2.20	3.2	2.14	0.4	2.09	−2.0
Be${}_{2}$	4.63	4.69	1.2	4.4	−5.0	4.15	−10.4	4.14	−10.7	4.14	−10.7
B${}_{2}$	3.00	3.08	2.6	–	–	–	–	3.16	5.3	3.01	0.0
C${}_{2}$	2.35	2.66	13.1	–	–	–	–	2.64	12.4	2.49	6.2
NO	2.17	2.19	0.7	–	–	–	–	2.02	−7.0	1.96	−9.9
CN	2.21	2.23	0.5	–	–	–	–	2.42	9.3	2.28	3.1
BeO	2.52	2.54	1.2	–	–	–	–	2.53	0.6	2.52	0.3
MAPE			2.4		24.9		10.8		8.2		6.0

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
