# Peer review of "Equilibrium Bond Lengths from Orbital-Free Density Functional Theory"

_molecules, 2020, doi:10.3390/molecules25081771_

Round 1
Reviewer 1 Report
Summary:
-------
The author presents the "second order atomic fragment approach" (2OAFA) as an implementation of orbital-free density functional theory (OF-DFT) based on the bifunctional formulation, where both the density and the potential are treated as independent variables in the functional. The 0OAFA of the author is to use the sums of densities and potentials for atoms to evaluate the energy functional of the molecule. For the 1OAFA, only the Pauli potential of a molecule is approximated with Pauli potentials of the atoms. In the manuscript the 2OAFA is presented for which, as the author states, "the molecular Pauli potential is approximated by the superposition of closed-shell atomic fragments...". Four example molecules are calculated, the Pauli potentials are discussed, and it is concluded that the 2OAFA "yields a significantly better description of chemical bonding."
Critique:
---------
The development of good functionals for OF-DFT is an important goal. The presented research is an original and very relevant step towards this goal and, as such, is highly important to be published. The journal is adequate and researchers interested in the development of orbital-free DFT will certainly want to read what has been tried by the author and what the results are.
Unfortunately, there are some issues with the presentation and the manuscript can, from my point of view, not be published in the present form. I think it is very much in the interest of the scientific community and in the author's interest to fix these issues before the manuscript is distributed. I identified a major issues that the author has to fix, a few issues that the author should fix, and a few minor issues that the author should consider. These are listed below.
Major issue:
------------
* The author claims that the 2OAFA "considerably improves" bond lengths compared to the 1OAFA. However, only 4 molecules are investigated and, from table 1, we see that the claim is true only for *one* molecule, N2. For the others, no significant change can be seen. Hence, the claim is not supported by the data.
The possible solutions of this problem are to support the claim by getting more data or to change the claim. If the author chooses the former path, I suggest to calculate a decent number of diatomics to get a reasonable picture of what is really happening. If the author chooses the latter path, I suggest the author focuses on explaining why the bond distance becomes much better for N2 but not for the other molecules. Also, I strongly suggest to remove the discussion of the "mean absolute relative error" if only four molecules are considered, as it makes little sense for such a small dataset and is currently even misleading.
BTW, if the author decides to get more data by calculating more molecules, I would suggest to rename "mean absolute relative error" to, e.g., "mean absolute percentage error", which is what a quick internet search provides as suggestion. I guess it was called this way because an error that is both absolute and relative sounds weird.
Issues:
-------
* The author states that in the 2OAFA, closed-shell atomic fragments are used. However, the explanation of the method consists of two sentences (starting at lines 76 and 115, respectively) and section 4. Closed shell means that two electrons occupy every spatial orbital. All atoms of the considered molecules O2, CO, and Be2 have an even number of electrons and are thus all closed-shell -- for those, what is the difference between the 1OAFA and the 2OAFA? Not much, as the numbers indicate, but I don't see a reason why 1OAFA and 2OAFA should be different at all. For N2, the nitrogen atom has an odd number of electrons. What is the closed-shell variant? The anion, the cation, a superposition? I'm a bit confused and the text in section 4 does not really clarify the chemical/physical/mathematical picture. Could this please be explained, especially what is meant with 'the atomic valence electron density has been restricted to closed-shell states' (line 204f)? Are 'states' maybe Slater determinants? Is this a CI calculation?
* The author states that relaxation effects induced by the molecular environment are taken into account. However, if I understood the method correctly, the "relaxation effect" is to use closed-shell atomic Pauli potentials instead of the Pauli potentials of the atomic ground state.
Relaxation due to the environment sounds like an actual optimization is done in the presence of the environment, and not like an ad hoc replacement of one atomic quantity with another. I would recommend that the author adjusts title and discussion to make clear what the motivation is compared to what is actually being done. In this respect, I also find the last paragraph (starting line 227) problematic, because the interpretation given there is the motivation but not (strictly) the conclusion from the results of the study.
Minor issues:
-------------
* The name "second order atomic fragment approach" indicates that there is some systematic improvement happening, but I don't see what a logical 3rd, 4th, ... order approach could be. For me it looks more like another variant of the 1OAFA. Maybe the author could think of a more appropriate name that does not have this problem, e.g. '1OAFA variant 2' or 'modified 1OAFA'?
* There are many problems with the formulations in the text, for example:
- line 14: 'reliable physical description' of what?
- line 34: The energy functional corresponding to what is to be approximated?
- line 47: 'Hereby' should be 'Here'. I think all occurrences of 'hereby' in the text make little sense.
- after line 49: 'expressed in from of an'
- after line 50: 'whereby' might be ok, but I would replace it with 'where' (also at other places)
- after line 59: 'Of course, the Pauli potential' Why 'of course', is it really self-evident? The phi_i are not molecular eigenfunctions (solutions of the many-electron problem) but Kohn-Sham orbitals, aren't they? And the eigenvalues epsilon_i are also the Kohn-Sham eigenvalues, right?
- line 66: 'of its atomic' -> 'of their atomic'
- line 88: 'are added after each data column' sounds strange, maybe 'are also given in the table'
- line 91: 'to its corresponding' -> 'to their corresponding'
- line 101: Please check the meaning of 'recompiled'.
- line 107: 'geometry' -> 'structure' or 'nuclear configuration' (occurs multiple times, e.g. line 132)
- line 127: 'the the'
- Table 1, caption: The sentence starting with 'Zeroth order level' might like to become a real sentence.
- line 134: 'from first' -> 'from the first'
- line 141: 'has significantly high values' -> e.g. 'is relevant'
- line 144: 'of' -> 'off'
- line 147: 'inspect'
- line 155: 'compared'
- line 167: 'waiting'
- line 175: 'decrease of values' Which values? Decrease compared to what?
- line 187: 'later' I don't see where time comes into play.
- line 198: What is meant with 'wavefunction mechanics'? Quantum mechanics? Wavefunction-based electronic structure theory?
- Figure 1: Why are the cuts (e), (f), and (g) shown? They are not discussed at all, aren't they? What is the unit of the colormap?
- line 203: What is meant with the shell concept being evaluated?
- line 210: 'the density, the Pauli' -> 'the density. The Pauli'
Several times, the word 'meaningful' is used. However, it is a bit meaningless as it is not explained what is actually meant with it. Maybe the author wants to remove it.
- line 337: The volume is 34, not 334.
Last words:
-----------
This might sound like I am very critical regarding the manuscript, but, as stated above, I am strongly convinced that the study is highly valuable and should be published, albeit not the manuscript as it presently is.
Author Response
Mayor issues:
1) Data has been evaluated form a larger number of dimers. Results are presented in Table 1. Additionally, the difference between the first order variant and the present variant with groundstate versus closed-shell Pauli potential was investigated. Numerous changes related to that major issue have been introduced in the manuscript.
MARE has been changed to MAPE.
Issues:
1) Atoms in their groundstate usually have a high-spin configuration due to the Pauli principle. However, artificial closed-shell atoms seem to be the better choice for atomic fragments. The differences have been highlighted in the revised version, e.g:
{\color{red} The largest influence of the electronic state of the atomic fragments is on the atomic Pauli potentials and as such on the
fragment Pauli potential of the molecule.
A much smaller indirect influence comes via the molecular density, which is also build from the atomic fragments,
see section~\ref{sec:Mat_Met} for further details.
While the electron density itself is not altered by the electronic fragment state, there is a slight influence on the total energy,
due to those energy parts that are treated in spin-polarized formalism.}
In this work, {\color{red} the influence of the electronic fragment states on the equilibrium bond length is studied in detail.
In the present work the molecular electron density is build from closed-shell fragments
(with equal number of alpha and beta electrons, leading to fractional occupations for uneven number of total electrons in the fragment),
while the influence of the Pauli potential is explored systematically
by taking either the atomic Pauli potential (PP) from atoms in their groundstate (gs)
or from atoms in an artificial closed-shell configuration (cs)....
2) The title has been changed to "Equilibrium bond lengths from orbital-free density functional theory"
The terminus "second order level" has been avoided throughout the manuscript.
Changes concerning this point (and previous points) invoke changes throughout the whole manuscript and are highlighted in red.
Minor issues:
1) line 14: "of matter" has been introduced.
2) line 34: "corresponding" has been omitted.
3) line 47: "Hereby" has ben changed in "Here" (also at later occurence)
4) after line 49: "from" has been changed to "form"
5) line 50: "whereby" has been changed to "where" (also at other places)
6) "Of course" has been omitted. "molecular" replaced by "KS". Additionally, ""for the molecule of interest" has been introduced in order to clarify between entities of the molecule and entities of the fragments.
7) line 66: "its" has been changed to "their"
8) line 88: "are added after each data column" has been changed to "are also given in the table"
9) line 91: "to its corresponding"has been changed to "to their corresponding"
10) line 101: "recompiled" has been changed to "compiled"
11) line 107: "geometry" has been changed to "structure" ( also at other places)
12) line 127: "the the " has been changed to "the"
13) Table 1: The sentence has been changed to
{\color{red} The zeroth order level has been performed with frozen fragments.
Equlibrium bond lenght from first order level and the recent work are evaluated
at valence optimized level\cite{Finzel-2019}. Bond distance from KS data are obtained with ADF \cite{ADF17} using LDA(Xonly) level with the QZ4P basis sets.}
14) line 134: "the" has been introduced
15) line 141: "has significantly high values" has been changed to "'is relevant"
16) line 144: "of" has been changed to "off"
17) line 147: "inspect" has been changed to "cf."
18) line 155: "compared" has been changed to "cf."
19) line 167: "waiting" has been changed to "weighting"
20) line 175: "decrease of values" has been changed to "{\color{red} This results} in a slight decrease of values {\color{red} for the kinetic energy}"
21) line 187: The referee is right. "later" has been changed to "at small internuclear distances"
22) line 198: "wavefunction mechanics" has been changed to "wavefunction-based electronic structure theory"
23) Figures e), f), and g) are now properly referenced in the text.
{\color{red} All orthoslices, shown in Figures~\ref{fig:N2_prom_cs_vp}, \ref{fig:N2_mol_vp}, and \ref{fig:N2_prom_gs_vp},
display almost spherical behavior around the atomic nuclei and fall off rapidly.
As a consequence the Pauli potentials are similar even in the bonding region.}
24) line 203: "being evaluated" has been changed to " are determined"
25) line 210: "the density, the Pauli" has been changed to "the density. The Pauli"
26) the reference has been corrected.
Reviewer 2 Report
This is a very interesting and well-written paper. I recommend that it be published after a couple of changes are made. In the fifth line of the Abstract, I suggest that "Pauli" be inserted between "the" and "potential".
Eq.(8) was actually first given in reference [ 37]. ( See Eq.(36) within this reference,). Consequently, reference [ 37 ] should be placed directly above Eq.(8), along with [ 48, 51 ].
If it is easy to determine, it would be nice to know if the approximate Pauli potential is non-negative everywhere, which is a property satisfied by the exact Pauli potential. ( I consider this an optional addition. ).
Author Response
1) "Pauli" was inserted at the requested place in the abstract line 5.
2) Ref. 37 was inserted above eq. 8.
3) A comment for the non-negativity was added.
{\color{red}Furthermore, the atomic fragment approach captures - by construction - important characteristics of the
Pauli potential, e.g. it maintains the atomic shell structure in the core region and is non-negative everywhere in space.
The fragment Pauli potential is constructed from the sum of the individual atomic KS Pauli potentials which obey the non-negativity condition.
Consequently, the fragment Pauli potential is non-negative everywhere.
The non-negativety of the Pauli potential has been shown to be an important aspect for the design of
orbital-free approximations for the kinetic energy \cite{Karasiev-Jones-Trickey-Harris-2009-prop, Karasiev-Chakraborty-Trickey-2013, Karasiev-Trickey-2015}.}
Reviewer 3 Report
Finzel's manuscript is a continuation of the paper “The first order atomic fragment approach – an orbital-free implementation of density functional theory” by the same author published in J. Chem. Phys. 151, 024109 (2019).
Energy curves for the four tested molecules, N2, O2, Be2 and CO, are presented in order to determine the respective equilibrium bond lengths. It is shown that the corresponding bond distances improve over the first order approach by reducing the mean absolute average error to 8.6%.
Author Response
Thank you very much for your time.
Round 2
Reviewer 1 Report
Thank you for the changes, in particular for testing further diatomics and for changing the title as well as a few parts of the text. It might well be that I do, from the text, now actually understand what your "new method" is, as explained below. However, I am still not 100% sure and would like you to clarify it, also in the interest of future readers.
In the review below, in part 1 I ask you for a clearer presentation of what you are doing. From my side this is the main point, and (under the condition that I understood what you are doing correctly) after you have done this, I have no major reservations and would not further delay the publication.
Thereafter, in part 2 I give you a recommendation for a change of style of the manuscript. I would do such a change because I think that it would improve the presentation, but it is additional work and also too much for me as a reviewer to ask you for that, as I think it is up to personal preference. Please feel free to ignore this recommendation.
In part 3 I give some specific recommendations regarding the text.
Part 1:
-------
From eqs. (12) and (13) I see that in 1OAFA you use the Pauli potential of the fragments (i.e., it is not changed during the energy optimization) but you treat the Pauli kinetic energy as functional of the density. In your article, what do you do? From lines 85ff I conclude that you use the sum of closed-shell atomic densities as approximation for the molecular density, and you use the sum of ground-state or closed-shell atomic Pauli potentials for the molecular Pauli potential. That's good, in this way you test the influence of the closed-shell idea on the bond lengths and, if this is really what you do, I even understand why there is a difference between the 1OAFA and the gs-PP data for O2, Co, and Be2 in your table (because you use the same Pauli potential, but a different electron density). However, this contradicts to some extend the statement in line 243, where you say you allow for a relaxation of the density (you actually allow less for a relaxation of the density than in the 1OAFA, don't you?).
In any case, I strongly suggest that you explain what you do with one or two simple equations, like eqs. (12) and (13). In this way the reader can immediately understand what you are actually calculating and comparing, and I can finally find out if my guess of what you are doing is correct.
Part 2:
-------
Also, if I am correct with my guess about what you are doing, I think your paper is actually not about presenting a "new method" but is a test the closed-shell idea. That is great and I think that your paper would be much stronger if you would present the discussion as such consistently. Your revisions were already along these lines but you might consider further changes in this direction. However, as stated above this is a personal recommendation that you are free to fully ignore.
To illustrate what I mean with my statement, I re-wrote the abstract accordingly:
"This work presents an investigation to model chemical bonding in various dimers based on the atomic fragment approach. The atomic fragment approach is an ab-initio, parameter-free implementation of orbital-free density functional theory which is based on the bifuntional formalism, i.e., it uses both the density and the Pauli potential as two separate variables. While providing the exact Kohn-Sham Pauli kinetic energy when the orbital-based Kohn-Sham data is used, the bifunctional formalism allows for approximations of the functional derivative which are orbital-free. In its first implementation, the atomic fragment approach uses atoms in their ground state to model the Pauli potential. Here, it is tested how artificial closed-shell fragments with non-integer electron occupation perform regarding the prediction of bond lengths of diatomics. Such fragments can sometimes mimic the electronic structure of a molecule better than ground-state fragments. It is found that bond lengths may indeed be considerably improved in some of the tested diatomics, in accord with predictions based on the electronic structure."
Part 3:
-------
21: 'big' -> 'considerable'
53+2: 'as been constructed' -> 'as being constructed'
69: 'and that all' -> 'and all' (maybe)
71: 'quantum mechanic package' -> 'quantum chemistry program' or 'electronic structure program'
80ff: I am not sure if you added/changed this text due to my previous request for clarification. Due to this part I think I understood what you are actually doing, but I had to read it multiple times and had to add some guess work. Can you please add the appropriate equations to this text, so that whatever you are doing is clearly stated?
112: I would call it "mean absolute percentage error" as used in other literature.
113: 'colorred'?
115: 'an average mean error' -> 'a MAPE'
122f: 'mean average percental error' -> 'MAPE'
147: 'tested molecules.' -> 'tested molecules compared to a MAPE of 8.2% when ground-state fragments are used.'
148: 'notabene' -> 'in particular' (maybe)
144: 'mean average percental error (MAPE)' -> 'MAPE'
Table 1: 'percental' -> 'percentage'
page 7/8: I like your discussion of the potentials.
199: 'of the order of magnitude' -> 'similar in magnitude'
209f: 'By the way, the figure nicely shows' -> 'The figure also illustrates'
234: Maybe I don't understand this, but can the "Shell concept" be determined according to the Slater rules? Or is that the effective quantum number which is determined according to the Slater rules?
243: As stated above: In this work the density is not updated in the Pauli kinetic energy (as it is in the 1OAFA) but is that of the closed-shell fragments, or? The statement 'The current approach allows for the relaxation of the density' is confusing me a bit.
257: 'resultant' -> 'resulting'
260: 'mean absolute percental error' -> 'MAPE'
260: This statement is, in my opinion, not strictly correct. You are comparing to the 1OAFA and you say that your MAPE is now better, but you have only calculated four molecules for the 1OAFA, hence you cannot compare the numbers. A correct statement would e.g. be 'It was shown that the corresponding bond distances improve for most diatomics if closed-shell fragments are used instead of ground-state fragments, with considerable improvements for some of the considered molecules.'
Also, I would not compare it to the 1OAFA but just say that using closed-shell fragments in the framework of the 1OAFA may be a good idea.
265: 'chemical bonding.' -> 'chemical bonding for some of the investigated diatomics.
Author Response
Part 1) The relaxation of the electron density is done in the first order approach and the present work by energetically optimizing the Slater exponent of the function governing the valence region (here 2S). The main difference comes form the atomic fragments, from which density and potentials are build. Those occupations are not altered during the optimization process. While they do not alter the density, they slightly influence the energetics (a comment to this is already present in the manuscript). Energy optimization by adjusting \alpha_2S is done in both methods, however with slightly different energy curves.
A comment for the optimization has been inserted in the theory section:
{\color{green} The atomic fragment approach allows for a relaxation of the electron density
expressed as the sum over nodeless, atom-centered Slater functions, see section~\ref{sec:Mat_Met} for details,
by adjusting the exponents of the Slater functions, while keeping their occupations fixed.}
In the section Materials and Methods a comment was added:
{\color{green}
\begin{equation}
\label{eq:rho}
\rho(\rv) = \sum_i g_i \phi_i^2(\rv - \Rv_A) \; \; .
\end{equation}
%
where the occupations $g_i$ correspond to the total number of electrons in the corresponding shell.}
I hope, that the method is now described clearly. The difference is in the fragments only, not in the optimization procedure.
------------------------------------------
Part 2) The abstract was changed accordingly. Additionally, the statement concerning the comparison between first order and cs versus gs fragments was changed in the conclusions as requested in part 3, together with the following statement:
{\color{green} Within the atomic fragemnts approach, the molecular Pauli potential
is approximated by the superposition of atomic Pauli potentials.
While in previous attempts only atomic groundstate fragemnts have been employed,
this work exploits the influence of the atomic fragment states on the ability to model chemical bonding.
Thus, in }
{\color{red} the present work} the molecular Pauli potential is approximated by the superposition of
closed-shell {\color{green} and groundstate} atomic Pauli potentials {\color{green}, respectively, }
and the electron density is determined by energy minimization.
-------------------------------------------
Part 3)
1) 21: 'big' has been changed to 'considerable'
2) 53+2: 'as been constructed' has been changed to 'as being constructed'
3) 69: 'and that all' has been changed to 'and all'
4) 71: 'quantum mechanic package' -has been changed to 'electronic structure program'
5) 80ff: A comment has been added in the theory section as well as additional changes in the section Materials and Methods, see also answer to Part 1.
6) 'percetntal' has been changed to 'percentage'
7) 113:missing \ was inserted
8) 115: 'an average mean error' has been changed to 'a MAPE'
9) 122f: 'mean average percental error' has been changed to 'MAPE'
10) 147: 'tested molecules.' has been changed to 'tested molecules compared to a MAPE of 8.2% when groundstate fragments are used.'
11) 148: 'notabene' has been changed to 'in particular'
12) 144: 'mean average percental error (MAPE)' -> 'MAPE'
13) Table 1: 'percental' has been changed to 'percentage'
14) 199: 'of the order of magnitude' has been changed to 'similar in magnitude'
15) 209f: 'By the way, the figure nicely shows'has been changed to 'The figure also illustrates'
16) 234: It basically means that s- and p- electrons are grouped, while d and f (not treated in the manuscript) are treated as separate shells. A short comment has been added: {\color{green}, e.g. second row elemets have one shell in the core region and one single shell in the valence region
(grouping s- and p-electrons together).}
17) 243: A comment has been added:
{\color{green} by adjusting the exponents $\alpha_i$}.
18) 257: 'resultant' has been changed to 'resulting'
19) 260: 'mean absolute percental error' has been changed to 'MAPE'
20) 260: statement changed accordingly:
{\color{green} It was shown that the corresponding bond distances improve for most diatomics
if closed-shell fragments are used instead of ground-state fragments,
with considerable improvements for some of the considered molecules}
by reducing the {\color{green} MAPE} to 6\%.
21) 265: 'chemical bonding.' has been changed to 'chemical bonding for some of the investigated diatomics.'
Round 3
Reviewer 1 Report
Thank you for the clarification and the changes. I spotted that line 269 now sounds a bit strange and you might want to change it, e.g. to
'by reducing the MAPE to 6%' -> 'and with a MAPE of only 6%'
Otherwise I have no more relevant comments.